# Match Algorithms for Scientific Names in FlorItaly, the Portal to the Flora of Italy

**DOI:** 10.3390/plants10050974

**Published:** 2021-05-13

**Authors:** Matteo Conti, Pier Luigi Nimis, Stefano Martellos

**Affiliations:** Department of Life Sciences, University of Trieste, via L. Giorgieri 10, I-34127 Trieste, Italy; matt.ciao@gmail.com (M.C.); nimis@units.it (P.L.N.)

**Keywords:** checklist, database, nomenclature, synonym, taxon

## Abstract

Scientific names are not part of everyday language in any modern country, and their input as strings in a query system can be easily associated with typographical errors. While globally unique identifiers univocally address a taxon name, they can hardly be used for querying a database manually. Thus, matching algorithms are often used to overcome misspelled names in query systems in several data repositories worldwide. In order to improve users’ experience in the use of FlorItaly, the Portal to the Flora of Italy, a near match algorithm to resolve misspelled scientific names has been integrated in the query systems. In addition, a novel tool in FlorItaly, capable of rapidly aligning any list of names to the nomenclatural backbone provided by the national checklists, has been developed. This manuscript aims at describing the potential of these new tools.

## 1. Introduction

*Nomina si nescis, perit et cognitio rerum*. If we ignore the names, we also lose the knowledge of things. This sentence by Linnaeus [1] highlights the relevance of names in biology (as in any human interaction). Scientific names provide access to a wealth of information collected during centuries of research [2]. For accessing the information on a taxon, it is necessary to know its currently accepted name. However, since any binomial—via the genus name—expresses a hypothesis on the phylogenetic affinities of a given taxon, scientific names do change over time, as our knowledge progresses [3,4,5]. While globally unique identifiers (GUIDs [6]), as the LSIDs [7] can be used to univocally address a taxon name, are useful in web services which allow machine–machine communication, they can hardly be used for querying a database manually. On the other hand, the use of taxon names for querying a database system could lead to several critical issues [8,9]. Among them, synonymies were once a problem, which is, however, at least partly addressed by means of nomenclatural thesauri, which can be invoked for retrieving the accepted name and all its synonyms (e.g., The Catalogue of Life [10]). Today, all modern online checklists have their own nomenclatural thesaurus or are connected to a web service or tool which can at least resolve synonyms [11]. However, another major issue is due to the fact that a taxon name is a string of text (in Latin), so that typing it into a query system can easily lead to typographical errors. Such an issue is normally addressed by means of matching algorithms (e.g., Taxamatch [12]), which find the closest match to any input string with one or more typos.
Match algorithms applied on taxon names are normally made of two components:A parser, which divides the string into its components (i.e., genus, species, authority, etc.). Some relevant examples are: gnparser [13], the name parser of GBIF [14], and the REGEXP parser [15].A matcher, which compares the output of the parser with a thesaurus of names. Among matchers, some are based upon phonetic similarity (such as Soundex [16]), while others use metrics for measuring orthographic differences among strings, such as the Levenshtein distance. An interesting example is the Taxamatch algorithm [17], which combines the two approaches.

Such algorithms are currently used in several data repositories, such as the GBIF (parser: GBIF name parser, matcher: Taxamatch), the iPlant Taxonomic Name Resolution Service (parser: gnparser, matcher: a modified version of Taxamatch), Euro+Med (parser: homebrew, matcher: a modified version of Taxamatch), WoRMS (parser: gnparser, matcher: a modified version of Taxamatch), and BiOnym (parser: REGEXP, matcher: several combined algorithms [15]).

As far as the vascular flora of Italy is concerned, FlorItaly, the Portal to the Flora of Italy (http://dryades.units.it/floritaly; accessed on 29 March 2021) [18], is built on the basis of the checklists by Bartolucci et al. [19] and Galasso et al. [20] and their subsequent updates, which are published twice per year [21,22,23,24,25,26,27,28,29,30,31,32]. FlorItaly hosts a rich thesaurus of names, and each query is parsed through it in order to provide all relevant results. In order to provide further help to users, it was decided to enrich FlorItaly with a novel tool, capable of retrieving a list of accepted names for any list of scientific names (input as a plain text or CSV file), deriving, e.g., from old literature, phytosociological relevés, lists of herbarium samples, image archives, or any existing database containing the names of plants occurring in Italy. This new tool, that will permit to rapidly align any list of names to the nomenclatural backbone provided by the national checklists, was envisaged as a relevant step towards the aggregation into FlorItaly of other resources, such as digitized herbaria, image archives, databases of vegetation reléves, etc. A near match algorithm was also adopted to support the query system of FlorItaly in order to resolve misspelled scientific names. This algorithm was included in the three query systems of FlorItaly (basic, standard, and advanced), which now warn users if a misspelled name was input and provide results for the closest matching string in the thesaurus of scientific names.

This paper aims at describing the algorithms which have been integrated into the Portal to the Flora of Italy, detailing their logic, aims, and uses.

## 2. Materials and Methods

All software was written in PHP 5.6 (but is compatible with PHP 7.x) and operates on a MySQL database. The reference database (as on 28th February 2021) contains 10,928 accepted names and 13,300 synonyms. All names in the reference database are written as whole strings, i.e., they are not split into elements (genus, species, authority, subspecies, etc.).

### 2.1. Data Preparation

Accepted names and synonyms are used in their original form (a single string) by the algorithm integrated in the query interfaces of FlorItaly (see below), while they are used in a parsed form by the algorithm which performs list matches. Each time the checklist is updated (see Martellos et al. [11]), all names are automatically parsed and stored in their parsed form in a data table, together with a phonetic equivalent for each epithet (generated by using a modified version of an algorithm developed by Reees in 2007 and included into Taxamatch [17]). As an example, the name *Huperzia selago* (L.) Bernh. ex Schrank & Mart. subsp. *selago* is parsed into the following components: *Huperzia* (genus), HIPIRSA (genus, phonetical), *selago* (species), SILAGA (species, phonetical), L. Bernh. ex Schrank & Mart. (species authority), subs. (infraspecific rank) *selago* (infraspecific rank), and SILAGA (infraspecific rank, phonetical).

### 2.2. The Algorithms

Two algorithms were developed, aiming at matching an input string or a list of taxa to the reference database, either returning the closest match (in the case of the algorithm integrated in the three query interfaces) or a list of accepted names (in the case of the algorithm operating on lists of names).

#### 2.2.1. Input String

The algorithm that is integrated in the query interfaces operates by comparing the input string to accepted names and synonyms in the reference checklist. If no exact match is returned, it continues by searching for the closest match, i.e., the string in accepted names or synonyms which is closer to the input string in terms of the minimum number of substitutions, removals, or insertions. This algorithm operates by two major constraints: (1) it allows matching strings which are longer or shorter than the original one by 1 character, i.e., with one insertion or removal only, and (2) it never inserts a blank space, nor replaces a blank space in the input string.

#### 2.2.2. Lists of Taxa

The algorithm operating on a list of taxa follows four steps: normalization, pre-processing, parsing, and matching.

##### (1) Normalization

In the normalization step, the input string is elaborated as follows: (a) special characters (question marks, exclamation marks, hashtags, underscores, quotes, and double quotes) are removed; (b) numbers—if present—are removed (the reference database does not contain years associated to authorities); (c) letters with diacritical marks are replaced with their ASCII equivalents; (d) a blank space (if not present) is added after each dot; (e) indications of uncertain identification such as “cf.” and “aff.” are removed; and (f) double blank spaces, along with leading and trailing blank spaces, are removed.

##### (2) Pre-Processing

In the pre-processing step, the normalized string is compared to the reference database. If there is an exact match, it is returned with a maximum score (100%).

##### (3) Parsing

In the parsing step, each element of the input string is identified, separated, and classified. As an example, the string “*Achillea millefolium* L. subsp. *sudetica* (Opiz) Oborny” is parsed into genus (“*Achillea*”), species epithet (“*millefolium*”), species authority (“L.”), infraspecific rank (“subsp.”), infraspecific epithet (“*sudetica*”), and infraspecific epithet authority (“(Opiz) Oborny”). The algorithm makes use of a set of rules that exploit the properties of PHP arrays, along with REGEX (Regular Expressions), to identify the different elements of the input string. The algorithm is case-sensitive and takes into account the combination of uppercase and lowercase letters to correctly identify the elements of a taxon name. However, it can operate in case-insensitive mode if the input string is entirely uppercase or lowercase or when “force case insensitive parsing” is selected among the advanced options by the user.

To cope with issues related to the presence in the authors’ names of strings which are the same as those used as infraspecific rank indicators, a set of REGEX rules were implemented. If even this verification process fails, the ambiguous string (“v.”, “f.”, “s.”, or “c.”) is considered as the rank indicator of the infraspecific taxon. Thus, the result of the parsing algorithm will be flagged as potentially wrong, and another verification step is applied after the matching step (see below).

##### (4) Matching

In this step, the parsed string is matched against the parsed reference database using a home-brew version of Taxamatch, which makes use of a modified Damerau–Levenshtein distance algorithm, together with a phonetic algorithm. The Levenshtein distance algorithm calculates the minimum number of deletions, insertions, or substitutions of a single character needed to transform a string into a second string, returning an edit distance value (ED). As an example, “*milefolius*” and “*millefolium*” have an ED of 2 (one insertion and one substitution). The modified Damerau–Levenshtein distance (MDLD) allows even multiple character transpositions at the cost of the number of transposed characters. As an example, “*canadsiens*” and “*canadensis*” have an ED of 2 (two characters’ transposition) instead of 4 (four characters’ substitution) as per the Levenshtein distance algorithm. Then, the phonetic algorithm transforms the input string elaborated by the MDLD algorithm into a simplified phonetic equivalent. If this matches the phonetic equivalent of one name in the reference database, they are considered as phonetic matches. Thus, every parsed element of the input string (except the authority and the infraspecific rank indicators) is matched against the reference database orthographically and phonetically [17]. To be considered as a match, a name in the reference database must pass at least one of the two tests.

Following the infraspecific epithet match, if the input name is flagged as potentially wrongly parsed (see (3) in Section 2.2.2), and the best match has an infraspecific epithet ED >= 2 (or there are no matches at the infraspecific epithet level), the algorithm foresees a further verification. If (a) the string leading to a parsing error is present in at least one of the taxon name authorities in the reference database which passed the species epithet match, and (b) both in the input string and in the taxon name authority of the reference database, the word following the one that can lead to a parsing error is the same, or begins with the same letter, and has an ED = 1 (calculated using the MDLD), then the string that can lead to a parsing error is considered as part of the authority, and the portion of the input string following the specific epithet is parsed for a second time. Otherwise, no changes are applied. Authors’ names are matched against the reference database using an *ngram* algorithm [17], returning a similarity score between 0 and 1.

In the matching step, a *ComponentScore* ranging from 0 to 1 is assigned to each parsed portion of the input string.
*ComponentScore* = 1 − (ED/maxED)(1)

ED is the editing distance calculated with the MDLD, while maxED is the maximum possible value of ED that is equal to the length of the longest of the two strings which are matched (the input and the reference).

As far as infraspecific epithets are concerned, the *ComponentScore* is decreased by 0.3 when the rank indicator is not the correct one. As an example, in the case of *Achillea millefolium* L. subsp. *millefolium*, if the input string is *Achillea millefolium* L. var. *millefolium*, the infraspecific taxon will receive a *ComponentScore* of 0.7 (1 − 0.3).

The final score of the “taxon name” portion of the input string (*TaxonScore*) is given by the average of the score of each name component. As far as authorities are concerned, another score (*AuthorScore*) ranging from 0 to 1 is calculated using an *ngram* algorithm [17].

The final score, called *ReturnedScore*, ranges from 0 to 100.
*ReturnedScore* = (*TaxonScore* × 0.9 + *AuthorScore* × 0.1) × 100(2)

##### (5) Algorithm Settings

Users can run the algorithm in standard or advanced mode. In standard mode, the algorithm always performs fuzzy matching. In addition, infraspecific epithets are recognized only when preceded by a commonly adopted rank indicator [33]. No issues arise if the rank indicator does not have a dot at the end or if it contains typos (as an example, to indicate the rank “subspecies”, the strings subsp., Subsp., and susp are all accepted).

Further settings can be adjusted by users: (a) selecting a custom infraspecific identifier for subspecies, variety, form, and cultivar, in the case that those used in the input dataset are not supported by the algorithm; (b) enabling or disabling the phonetic match algorithm; (c) selecting the threshold score (from 1 to 100) under which a result is not returned: if 100 is selected, only taxon names that are an exact match are returned; and (d) forcing the parsing algorithm to be case-insensitive (useful if the input names do not follow the ICBN uppercase and lowercase convention).

## 3. Results

The algorithm integrated in the query systems of FlorItaly aims at improving effectiveness and usability of the system. Each of the three query interfaces (basic, standard, and advanced) allows to combine a string in the taxon name with several parameters checked or selected from closed lists. As an example, in the advanced query interface, users can query for all taxa known to occur in the region Friuli Venezia Giulia as archaeophytes, whose name contains the string “*officinalis*”. The result is a list of two taxa, *Calendula officinalis* L. and *Galega officinalis* L. However, while typing a string, it is possible to input one or more typos, thus biasing the results of the query. As an example, if the input string is “*officialis*” (an “*n*” is missing, Figure 1), the result would be an empty list, and users could be led to the conclusion that the two taxa are not known to occur in the region. The new algorithm allows to overcome this error, finding the nearest match for the input string and then performing the query on it. Thus, the output (Figure 2) highlights that the query string has been replaced by a “corrected” string by the algorithm (“*officinalis*”), thus making users aware that the string they input was not present in the database.

The algorithm operating on lists of taxa aims at providing a novel service in FlorItaly, i.e., the possibility to retrieve a list of accepted names after input of a list of scientific names. Users can register to the system and store up to three lists of names in their workspace. Thus, they can operate on them in different moments for resolving uncertain matches. A list of names (e.g., deriving from the digitization of vegetation relevés taken from a scientific paper) can be input either as a text (each name in a new line) or as CSV (comma-separated values) files. In the latter case, users must specify, through a graphic interface, which column in the input file hosts the taxon names. Furthermore, users can:Select the character used as a separator (comma, colon, semicolon, etc.);Select the terms used for infraspecific taxonomic ranks, such as subspecies, variety, and form;Allow the inclusion/exclusion of phonetic matches;Increase the tolerance of the algorithm, i.e., accepting a wider range of near matches than those allowed by the default settings. Increasing the degrees of freedom of the algorithm increases the time required for performing the query.

After the upload, the system performs a SELECT DISTINCT query on the column hosting the names, thus removing duplications. The algorithm will then compare each name in the list with the thesaurus of FlorItaly and return a page (Figure 3) listing the number of names (a) in the original list, (b) with no match, (c) with unambiguous match, and (d) with ambiguous match. The time required for uploading and matching a list depends on the size of the input. Normally, it ranges from 30 s to 5 min. The upper limit in the input files is 5000 names. Users can hide/show any part of the list and can operate on ambiguous matches only, selecting a match among those proposed by the algorithm (each with the corresponding score). At the end of the process, users can either download a CSV file with four columns (original names, accepted names, synonyms—when the input name matches a synonym and not an accepted name—and the match score) or, if they are logged in, save and continue the process in another moment.

To test the effectiveness of this algorithm, three heterogeneous datasets in CSV format were used:Dataset A: 890 names, without authorities, and “s.” as the infraspecific identifier for subspecies;Dataset B: 2981 names, all written in uppercase;Dateset C: 304 names written following the current code of nomenclature (ICBN).The results are reported in Table 1.

## 4. Discussion

Scientific names written in Latin are not part of everyday language in any modern country. Thus, their input as strings in a query system can generate several issues [34] and can be associated with typos. In the authorities’ portion of the name, furthermore, different abbreviations or even the inclusion or exclusion of a blank space could lead to biases in the results of a query. The query system of FlorItaly was originally developed as an exact matcher, with no automated correction of the input string(s). However, in order to improve users’ experience, it was decided to switch from an exact to a near match algorithm. This new algorithm has been integrated in the three query systems of FlorItaly, which now warn users if a misspelled string was input and provide results for the string which is the closest match in the reference database. As an example, for retrieving the data for *Ailanthus altissima* Mill., a user can query FlorItaly for the string “*lanthus alt*”. The only name matching the query is *A. altissima*. However, if the input string was “*lantus alt*”, with an omission error, the old query system did not return any result. The novel algorithm, on the contrary, corrects the input string into “*lanthus alt*”, warns the user, and returns the scientific name in the database which is the nearest match, i.e., *A. altissima*.

Other than querying FlorItaly by taxon name, a new service is provided by the possibility of comparing lists with the reference database. The new service can be useful, especially in the framework of the digitization of previously published botanic data, as well as herbarium specimens—efforts which are being carried out by the botanic community in Italy and abroad. Since old publications, as well as specimen labels, normally do contain outdated nomenclature, a system for resolving all the names in a batch process can be a relevant time saver to researchers. The algorithms integrated in FlorItaly are based upon Taxamatch [17]. However, while Taxamatch matches a scientific name up to the species level, the algorithms in FlorItaly support up to two infraspecific epithets, following the TNRS approach [9,35]. Furthermore, the algorithms were developed to address some matching issues which occur in the TNRS [36], especially when: (a) input names are entirely in upper- or lowercase; (b) the word “f.” is present (it could be part of the author’s name—filius—or the taxonomic rank “form”); or (c) epithets of species, or infraspecific ranks, are input with uppercase initials. These issues are related to the fact that the principal parsing algorithms, *gnparser* and *gbif parser*, are case-sensitive and use the presence of upper- and lowercase characters to properly parse the input strings. This approach allows for an accuracy of 99% when input names are written following the nomenclatural code rules. However, when this is not the case, part of the accuracy is lost [13,37]. Another relevant issue occurs when parsing a scientific name in which part of the authority can be the same as an infraspecific rank indicator. In the case of names written following the ICBN uppercase and lowercase rules, this could happen for the string “f.” only, which can either refer to “filius” (part of the authority) or “form” (infraspecific rank). However, when names are not written following the ICBN uppercase and lowercase rules, as in the case of names written entirely in uppercase, “V.”, “F.”, “S.”, and “C.” can lead to incorrect parsing, since they can either be part of the authority or refer to infraspecific ranks (“variety”, “form”, “subspecies”, and “cultivar”). Note that “C.” and “S.” are treated as infraspecific rank indicators by the algorithms in FlorItaly only if they are manually selected by the user as infraspecific rank indicators, since they are not commonly used [33]. In order to properly address these issues, algorithms were developed mainly focusing on extending the already effective performance of Taxamatch [17] for handling infraspecific ranks. This is especially relevant in Italy, since lists of names—which can derive, e.g., from digitization efforts—derive from several older backbones, such as the floras by Fiori [38], Zangheri [39], or Pignatti [40], who made extensive use of infraspecific ranks that often are not written following the ICBN standard.

The inclusion into FlorItaly of the two algorithms aims at providing the scientific community with further services for improving data retrieval through the use of taxon names as identifiers. There is, especially in Italy, a wealth of different resources hosting digitized data on plants. Some of them are already aggregated, and their output is provided together with checklist data in the taxon pages of FlorItaly [18]. However, this was possible only because these resources share a common nomenclatural backbone. Thus, providing a simple, fast, and reliable tool for aligning any list of names to the nomenclatural backbone, provided by the national checklists, is a relevant step towards the aggregation of other resources, such as digitized herbaria, image archives, databases of vegetation reléves, etc. The aggregation of databases of phytochemical and ethnobotanic data is already ongoing, together with datasets of primary biodiversity data. The development of a web service to provide data from FlorItaly to other resources, such as the National Biodiversity Network of the Ministry of Environment, is one of the relevant developments planned for the future.

These developments, together with the incoming novel National Center for Botanic Data of the European Research Infrastructure LifeWatch, which will be based at the University of Bologna, aim at providing the scientific community with a unique entry point for botanic data concerning the vascular flora of Italy.

## Figures and Tables

**Figure 1 plants-10-00974-f001:**
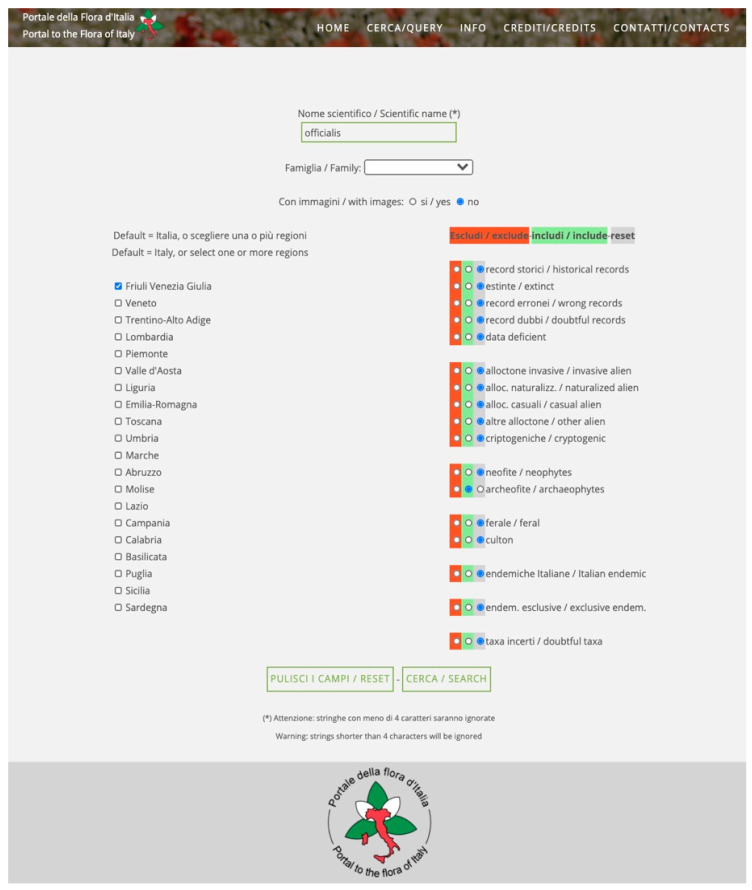
Results of a query in the advanced query interface of FlorItaly. The input name is incorrectly spelled (“*officialis*” instead of “*officinalis*”).

**Figure 2 plants-10-00974-f002:**
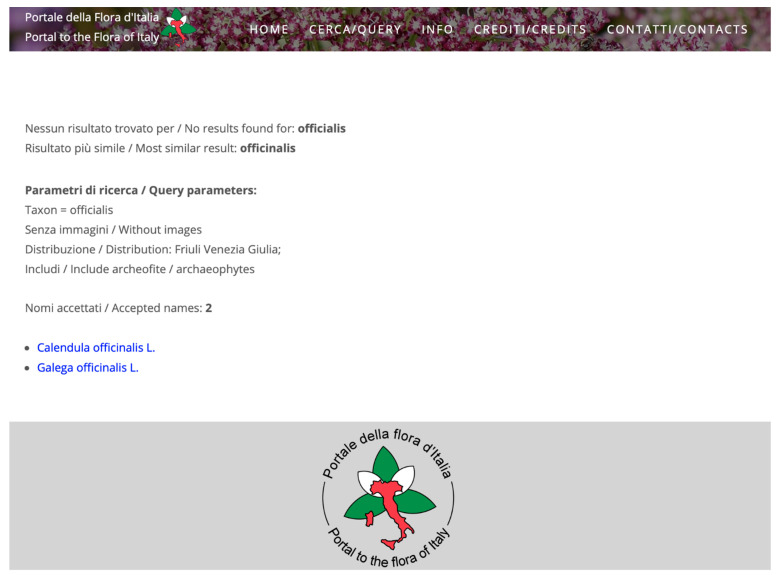
Advanced query result from FlorItaly. Since no exact matches were found for the misspelled input string “*officialis*”, the near match algorithm returned the most similar string in the database (“*officinali**s*”) and performed the query by using it. Users are provided with a warning at the beginning of the results page.

**Figure 3 plants-10-00974-f003:**
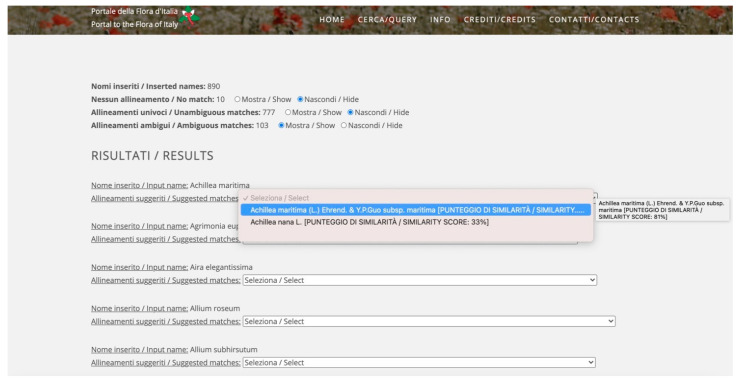
Result page after the input of a CSV file containing a list of names. The interface highlights the number of names in the list and the number of no matches, unambiguous, and ambiguous matches. Each part of the list can be hidden. As far as ambiguous matches are concerned, users can select a match among those proposed by the algorithm (each listed with its matching score). When strings are too long, holding the pointer on the incomplete string for a few seconds allows to view the full string in a text pop-up.

**Table 1 plants-10-00974-t001:** Results of the list matching algorithm on three different datasets (see text). For each dataset, other than the total number of taxa, positive matches (divided into unambiguous and ambiguous, the latter requiring user input) and no matches are reported.

Dataset	Total Names	Unambiguous Matches	Ambiguous Matches	No Matches
A	890	609 (68.43%)	271 (30.45%)	10 (1.12%)
B	2981	2649 (88.86%)	296 (9.93%)	36 (1.21%)
C	304	293 (96.38%)	10 (3.29%)	1 (0.33%)

The source code developed during this study is available on GitHub by querying for “FlorItaly name match”.

## Data Availability

All data are available online in FlorItaly (http://dryades.units.it/floritaly, accessed on 29 March 2021).

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
