# Peer review of "Match Algorithms for Scientific Names in FlorItaly, the Portal to the Flora of Italy"

_plants, 2021, doi:10.3390/plants10050974_

Round 1
Reviewer 1 Report
Dear Authors,
I carefully read your manuscript entitled "Match algorithms for scientific names in FlorItaly, the portal to the flora of Italy" and I found it very interesting and very useful for botanists and other scholars who don't have a sufficient botanical background to understand the differences among several and continuously updated scientific names. However, I believe you should better review the form of your manuscript: is it a research article? Or a communication or a note? Please check it out better.
Also, I believe that you should better take care of the background of your work. There are very few bibliographic references, especially in the "Introduction" section. A work like this, to better support the choices made, should also conduct many comparisons in the "Discussion" section. As already indicated in line 56, you should correct the reference [12] of the "References" section because it does not correspond to Bartolucci et al. (2018). Furthermore, you should add all subsequent updates of Bartolucci et al. (2018) and Galasso et al. (2018). I don't understand why you didn't: the reader will be greatly helped to understand better.
Other small notes are indicated in the attached PDF.
After all these improvements, in my opinion your manuscript could continue in its publication process.
Best wishes.

Author Response
Dear reviewer,
we are grateful for your useful comments to our manuscript.
To improve the manuscript's background, we added several citations to support some statements in the introduction section. Furthermore, as you requested, we cited all the 12 updates of the original checklists published since 2018. Plus, we corrected the reference to Bartolucci et al. (2018). We also tried to further improve the text. However, given the input from the other 4 reviewer, which do not require any modification, but some slight improvements in the text, we limited the changes to the minimum.
As far as the line "the strings subsp., subsp., and susp", there was an error, since it should have been "the strings subsp., Subsp., and susp". We have corrected it in the manuscript.
As far as your comment on Fig. 2 caption: "Please, try to explain better why the most similar string in the database is "officinali" instead of "officinalis". You can do it in the text, near the lines 202-204.", since the day we submitted the manuscript, the ‘fuzzy match’ algorithm which was constrained to return strings of the same length of the input string was modified and improved. Thus, while at that time the most similar to the input (officialis) was a same length string (10 characters, officinali), now it can return as the closest match strings of different length than the input (+- 1 character). Thus, we have changed figure 2 and its caption, as well as the text, since now the closest match is, as you suggested, officinalis.
We confirm that this manuscript is a research paper, since it describes the development of novel algorithms, which build upon previous experiences in the field. We highlight this by adding a few lines in the discussion section.
Best regards,
on behalf of the authors,
Stefano Martellos
Reviewer 2 Report
The paper by Conti et al. presents a new tool to support plant scientific names macth and query, which is implemented in the Portal of the Flora of Italy. It facilitates the query of the database reducing the risk of mispelling the name, and allows the comparison of the current plant list with new datasets coming, for instance, from the digitalization of herbaria or phytosciological vegetation plots. The taxonomy of plant species is constantly updated, also based on new evidences brougth by molecular analyses, and tools such the one presented in this paper are very useful to keep track of the chahes. I reccommend the publication with only some minor changes of the English
Author Response
Dear reviewer,
Thanks for your comments.
We have reviewed the manuscript for improving the language.
Best regards,
on behalf of the authors,
Stefano Martellos
Reviewer 3 Report
Dear Authors,
The manuscript Match Algorithms for Scientific Names in FlorItaly, the Portal to the Flora of Italy describes clearly and easily the algorithms that further improve the performance of FlorItaly, making it an important tool for technicians and researchers involved in the botanical data management.
In my opinion the manuscript is suitable for publication with minor revisions. Few mistakes (mainly typing) and comments are reported in the attached file.

Author Response
Dear reviewer,
thanks for your comments.
We have checked the manuscript for typing errors, and improved some sentences.
Best regards,
on behalf of the authors,
Stefano Martellos
Reviewer 4 Report
This paper describes about the development of an algorithm that allows some errors and suggests the correct answer for scientific name searches, where input errors are likely to occur frequently. With the increasing availability of digital images of plant specimens and specimen information on the Internet around the world, there is a great demand for scientific name search programs that can perform such near-match searches. If the authors have the option of publishing the program of the near-matching algorithm for scientific names developed in this paper on Github or distributing the program on demand to those who wish to use it, the publication of the paper will be highly significant. We hope that the authors will positively consider releasing the program.
L48 GIBF --- GBIF
Author Response
Dear reviewer,
thank you for your comments.
we have published the code on Github. We added a line at the end of the results section reporting that it is findable by querying "FlorItaly name match".
Best regards,
on behalf of the authors,
Stefano Martellos
Reviewer 5 Report
The authors describe the changes that they made to improve the FlorItaly, the portal to the online flora of Italy. Precisely, the authors integrated a near-match algorithm to the existing portal that could resolve misspelled scientific names and added a novel tool that could allow rapidly align any species list to an existing nomenclatural backbone in the portal. This is a very interesting manuscript, very well written, and with almost no typographical errors. The methods are well presented. The results deviate from the normal biology papers, which can be predictable this type of manuscripts. The conclusions are sound and clear.
Very minor issues identified
Lines 54 and 57: where FlorItaly instead of “Floritaly”.
Line 112: add a space between “e)” and “indications”.
Line 240 - 241: “ …. the number of no, unambiguous, and ambiguous matches …”. Is “the number of no” correct?
Line 246 – 248: It’s not obvious to the reader that a), b) and c) correspond to A, B and C in table 1.
Line 253: Scientific names are written in Latin, are not part of everyday language in any modern country. Consider deleting “are”. “Scientific names written in Latin are not part of everyday…”
Author Response
Dear reviewer,
thank you for your comments. We have corrected the minor issues you highlighted.
The sentence in line 240 - 241: “ …. the number of no, unambiguous, and ambiguous matches …” has been changed into “... the number of no matches, unambiguous and ambiguous matches ...“.
The a), b) and c) in line 246 – 248 has been changed into A, B and C, according to the table.
Best regards,
on behalf of the authors,
Stefano Martellos
Round 2
Reviewer 1 Report
Dear Authors,
I can see that you modified the manuscript as suggested by the reviewers. Well done! In my opinion your manuscript can be accepted for publication in Plants journal.
Very compliments.